# BOLD and GenBank revisited – Do identification errors arise in the lab or in the sequence libraries?

Mikko Pentinsaari[1], Sujeevan Ratnasingham[1], Scott E. Miller[2], Paul D. N. Hebert[1]*

**1** Centre for Biodiversity Genomics, University of Guelph, Guelph, Ontario, Canada, **2** National Museum of Natural History, Smithsonian Institution, Washington, DC, United States of America

* phebert@uoguelph.ca

## Abstract

Applications of biological knowledge, such as forensics, often require the determination of biological materials to a species level. As such, DNA-based approaches to identification, particularly DNA barcoding, are attracting increased interest. The capacity of DNA barcodes to assign newly encountered specimens to a species relies upon access to informatics platforms, such as BOLD and GenBank, which host libraries of reference sequences and support the comparison of new sequences to them. As parameterization of these libraries expands, DNA barcoding has the potential to make valuable contributions in diverse applied contexts. However, a recent publication called for caution after finding that both platforms performed poorly in identifying specimens of 17 common insect species. This study follows up on this concern by asking if the misidentifications reflected problems in the reference libraries or in the query sequences used to test them. Because this reanalysis revealed that missteps in acquiring and analyzing the query sequences were responsible for most misidentifications, a workflow is described to minimize such errors in future investigations. The present study also revealed the limitations imposed by the lack of a polished species-level taxonomy for many groups. In such cases, applications can be strengthened by mapping the geographic distributions of sequence-based species proxies rather than waiting for the maturation of formal taxonomic systems based on morphology.

## Introduction

Species identifications play an important role in forensic analyses in contexts ranging from the interception of trade in CITES-listed species [1] to ascertaining the post mortem interval [2]. There are also expanding opportunities to track the movement of objects and organisms linked to their associated DNA. Although species identifications can play an important role in these contexts, the lack of taxonomic specialists often impedes analysis, a factor which has provoked interest in DNA-based approaches to species identification. Past studies have established that DNA barcodes can often assign specimens to their source species, but have also revealed differences in success among the kingdoms of eukaryotes. For example, the three barcode regions

**Data Availability Statement:** This study is a reanalysis of already published data that was submitted to GenBank. GenBank accessions are referenced in Table 1.

**Funding:** This work was enabled by Funding from the Canada First Research Excellence Fund, the Ontario Ministry of Research and Innovation, the Canada Foundation for Innovation, and Natural Sciences and Engineering Research Council of Canada. The funders had no role in study design, data collection and analysis, decision to publish, or preparation of the manuscript.

**Competing interests:** The authors have declared that no competing interests exist.

(rbcL, matK, ITS2) for plants deliver lower success than the single gene region (cytochrome *c* oxidase I, COI) used for animals [3]. Because COI generally has high accuracy in species assignment [4–9], the conclusions from a recent study by Meiklejohn et al. [10] were surprising. They assessed the capacity of reference sequences in BOLD, the Barcode of Life Data System [11], and GenBank [12] to generate species-level identifications. Their analysis revealed that both platforms performed similarly in identifying plants and macrofungi, but fared poorly in identifying insect species with BOLD showing lower success than GenBank (35% vs. 53%). They noted that their observed identification success did not conform to earlier results from DNA barcoding studies on insects, but did not carefully examine the possible causes of this unexpectedly low success. By evaluating the factors underpinning the incorrect assignments, the present study revealed that errors in sequence acquisition and interpretation accounted for most, if not all, of the misidentifications. To avoid similar issues in future studies, there is a need to adopt more rigorous procedures for data acquisition and analysis, and to reduce the current reliance on immature taxonomic systems.

## Material and methods

Meiklejohn et al. [10] analyzed 17 insects including representatives from 12 insect orders– Coleoptera (1), Dermaptera (1), Diptera (5), Ephemeroptera (1), Hymenoptera (1), Lepidoptera (2), Mecoptera (1), Neuroptera (1), Odonata (1), Orthoptera (1), Pthiraptera (1), and Siphonaptera (1). The specimens were obtained from the Smithsonian's National Museum of Natural History (USNM); most were collected 20+ years ago (e.g. *Pediculus humanus*– 1955). Following DNA extraction, the barcode region of COI was PCR amplified and then Sanger sequenced. Reflecting the DNA degradation typical of museum specimens, the sequences recovered were often incomplete (e.g. 254 bp for *Hexagenia limbata*). The resultant sequences were injected into the ID engine on BOLD [11] and into the BLAST function on GenBank [12]. This analysis delivered correct species identifications for six specimens (35%) on BOLD and for nine (53%) on GenBank.

The present study was initiated by downloading the 17 sequences from GenBank. These sequences were compiled into a dataset on BOLD (dx.doi.org/10.5883/DS-FBI2019), and specimen metadata unreported by Meiklejohn et al. were added to the sequences based on the labels of the voucher specimens deposited at the USNM, along with images of most specimens. The sequences were resubmitted to the BOLD ID engine and to GenBank BLAST with self matches excluded. Because some of the resultant identifications deviated from those reported in [10], the factors responsible for this discordance were examined.

## Results and discussion

### ID results from BOLD and GenBank

Table 1 compares the ID results for the 17 specimens between [10] and those obtained in the present study. The IDs from BLAST matched those reported by [10] as did ten of the IDs from BOLD. The other seven IDs from BOLD corresponded to those from GenBank, but not with the results in [10]. There was a simple explanation for this discordance. Meiklejohn et al. [10] had submitted the reverse complement rather than the coding sequence into the ID engine on BOLD, an approach which generated distant matches. Avoiding this misstep, the number of "correct" identifications generated by BOLD and GenBank was similar (12/17 at the genus level, 9/17 at the species level). In order to prevent further occurrence of such errors, the BOLD user interface has been updated to instruct users to enter COI barcode sequences in the ID engine in the forward orientation, and to warn users if the resulting identifications are suspected to result from reverse or reverse complement sequences.

**Table 1. Comparison of query results (top matches) for 17 insect species between Meiklejohn et al. [10] and the present study.**

| Query sequence | Order | Family | Genus | Species | Meiklejohn et al. | | | | | Present study | | | | |
|---|---|---|---|---|---|---|---|---|---|---|---|---|---|---|
| | | | | | Database | Sequence length | Top match genus | Top match species | Identity | Top match order | Top match family | Top match genus | Top match species | Identity |
| MK905407 | Coleoptera | Scarabaeidae | Phanaeus | vindex | GenBank | 581 | Phanaeus | sp. | 0.99821 | Coleoptera | Scarabaeidae | Phanaeus | sp. | 0.9982 |
| MK905407 | Coleoptera | Scarabaeidae | Phanaeus | vindex | BOLD_all | 581 | Phanaeus | sp. | 0.9982 | Coleoptera | Scarabaeidae | Phanaeus | sp. | 0.9982 |
| MK905402 | Dermaptera | Forficulidae | Forficula | auricularia | GenBank | 429 | Forficula | auricularia | 0.99299 | Dermaptera | Forficulidae | Forficula | aff. auricularia A | 0.9930 |
| MK905402 | Dermaptera | Forficulidae | Forficula | auricularia | BOLD_all | 429 | Dyscheralcis | retroflexa | 0.5 | Dermaptera | Forficulidae | Forficula | auricularia-A | 0.9976 |
| MK905402 / RC | Dermaptera | Forficulidae | Forficula | auricularia | BOLD_all | 429 | | | | Lepidoptera | Geometridae | Dyscheralcis | retroflexa | 0.5 |
| MK905396 | Diptera | Calliphoridae | Chrysomya | rufifacies | GenBank | 592 | Chrysomya | rufifacies | 1 | Diptera | Calliphoridae | Chrysomya | rufifacies | 1 |
| MK905396 | Diptera | Calliphoridae | Chrysomya | rufifacies | BOLD_all | 592 | Chrysomya | rufifacies | 1 | Diptera | Calliphoridae | Chrysomya | rufifacies | 1 |
| MK905397 | Diptera | Calliphoridae | Calliphora | vicina | GenBank | 553 | Calliphora | vicina | 0.98373 | Diptera | Calliphoridae | Calliphora | vicina | 0.9837 |
| MK905397 | Diptera | Calliphoridae | Calliphora | vicina | BOLD_all | 553 | Calliphora | vicina | 1 | Diptera | Calliphoridae | Calliphora | vicina | 1 |
| MK905393 | Diptera | Culicidae | Aedes | aegypti | GenBank | 658 | Aedes | aegypti | 0.99848 | Diptera | Culicidae | Aedes | aegypti | 0.9985 |
| MK905393 | Diptera | Culicidae | Aedes | aegypti | BOLD_all | 658 | Aedes | aegypti | 1 | Diptera | Culicidae | Aedes | aegypti | 1 |
| MK905403 | Diptera | Glossinidae | Glossina | palpalis | GenBank | 611 | Glossina | brevipalpis | 0.971 | Diptera | Glossinidae | Glossina | brevipalpis | 0.9710 |
| MK905403 | Diptera | Glossinidae | Glossina | palpalis | BOLD_all | 611 | Glossina | brevipalpis | 0.9694 | Diptera | Glossinidae | Glossina | brevipalpis | 0.9694 |
| MK905404 | Diptera | Muscidae | Musca | domestica | GenBank | 645 | Cryptopygus | tricuspis | 0.996 | Entomobryomorpha | Isotomidae | Cryptopygus | tricuspis | 0.9960 |
| MK905404 | Diptera | Muscidae | Musca | domestica | BOLD_all | 645 | Amphiura | incana | 0.5571 | Entomobryomorpha | Isotomidae | Folsomia | cf. diplopthalma | 1 |
| MK905404 / RC | Diptera | Muscidae | Musca | domestica | BOLD_all | 645 | | | | Ophiurida | Amphiuridae | Amphiura | incana | 0.5586 |
| MK905400 | Ephemeroptera | Ephemeridae | Hexagenia | limbata | GenBank | 254 | Glossina | brevipalpis | 0.9681 | Diptera | Glossinidae | Glossina | brevipalpis | 0.9681 |
| MK905400 | Ephemeroptera | Ephemeridae | Hexagenia | limbata | BOLD_all | 254 | Glossina | brevipalpis | 0.9675 | Diptera | Glossinidae | Glossina | brevipalpis | 0.9675 |
| MK905409 | Hymenoptera | Vespidae | Vespula | squamosa | GenBank | 560 | Vespula | squamosa | 0.99643 | Hymenoptera | Vespidae | Vespula | squamosa | 0.9964 |
| MK905409 | Hymenoptera | Vespidae | Vespula | squamosa | BOLD_all | 560 | Vespula | squamosa | 1 | Hymenoptera | Vespidae | Vespula | squamosa | 1 |
| MK905395 | Lepidoptera | Saturniidae | Callosamia | promethea | GenBank | 354 | Callosamia | promethea | 0.9969 | Lepidoptera | Saturniidae | Callosamia | promethea | 0.9969 |
| MK905395 | Lepidoptera | Saturniidae | Callosamia | promethea | BOLD_all | 354 | Callosamia | promethea | 0.9938 | Lepidoptera | Saturniidae | Callosamia | promethea | 0.9940 |
| MK905401 | Lepidoptera | Nymphalidae | Danaus | plexippus | GenBank | 623 | Danaus | plexippus | 1 | Lepidoptera | Nymphalidae | Danaus | plexippus | 1 |
| MK905401 | Lepidoptera | Nymphalidae | Danaus | plexippus | BOLD_all | 623 | Danaus | plexippus | 1 | Lepidoptera | Nymphalidae | Danaus | plexippus | 1 |
| MK905405 | Mecoptera | Meropeidae | Merope | tuber | GenBank | 655 | Merope | tuber | 0.92006 | Mecoptera | Meropeidae | Merope | tuber | 0.9201 |
| MK905405 | Mecoptera | Meropeidae | Merope | tuber | BOLD_all | 655 | Craesus | alniastri | 0.5 | Mecoptera | Meropeidae | Merope | tuber | 0.9430 |
| MK905405 / RC | Mecoptera | Meropeidae | Merope | tuber | BOLD_all | 655 | | | | Hymenoptera | Tenthredinidae | Craesus | alniastri | 0.5 |
| MK905408 | Neuroptera | Ascalaphidae | Ululodes | quadripunctatus | GenBank | 635 | Ululodes | quadrimaculatus | 1 | Neuroptera | Ascalaphidae | Ululodes | quadripunctatus | 1 |
| MK905408 | Neuroptera | Ascalaphidae | Ululodes | quadripunctatus | BOLD_all | 635 | Xanthopimpla | sp. | 0.5152 | Neuroptera | Ascalaphidae | Ululodes | quadripunctatus | 1 |
| MK905408 / RC | Neuroptera | Ascalaphidae | Ululodes | quadripunctatus | BOLD_all | 635 | | | | Hymenoptera | Ichneumonidae | Xanthopimpla | sp. | 0.5152 |
| MK905399 | Odonata | Gomphidae | Gomphus | exilis | GenBank | 612 | Cecidomyiidae | sp. | 0.9934 | Diptera | Cecidomyiidae | Cecidomyiidae | sp. | 0.9935 |
| MK905399 | Odonata | Gomphidae | Gomphus | exilis | BOLD_all | 612 | Dolichophis | schmidti | 0.5283 | Diptera | Cecidomyiidae | Cecidomyiidae | sp. | 0.9967 |
| MK905399 / RC | Odonata | Gomphidae | Gomphus | exilis | BOLD_all | 612 | | | | Squamata | Colubridae | Dolichophis | schmidti | 0.5283 |
| MK905398 | Orthoptera | Gryllidae | Gryllus | assimilis | GenBank | 278 | Gryllus | pennsylvanicus | 0.9964 | Orthoptera | Gryllidae | Gryllus | pennsylvanicus | 0.9964 |
| MK905398 | Orthoptera | Gryllidae | Gryllus | assimilis | BOLD_all | 278 | Gryllus | pennsylvanicus | 0.9964 | Orthoptera | Gryllidae | Gryllus | pennsylvanicus | 0.9964 |
| MK905394 | Siphonaptera | Pulicidae | Ctenocephalides | felis | GenBank | 643 | Pulex | irritans | 0.9642 | Siphonaptera | Pulicidae | Pulex | irritans | 0.9642 |
| MK905394 | Siphonaptera | Pulicidae | Ctenocephalides | felis | BOLD_all | 643 | Natrix | tessellata | 0.6176 | Siphonaptera | Pulicidae | Pulex | irritans | 0.9642 |
| MK905394 / RC | Siphonaptera | Pulicidae | Ctenocephalides | felis | BOLD_all | 643 | | | | Squamata | Colubridae | Natrix | tessellata | 0.6176 |
| MK905406 | Phthiraptera | Pediculidae | Pediculus | humanus capitis | GenBank | 384 | Stylops | sp. | 1 | Strepsiptera | Stylopidae | Stylops | sp. | 1 |
| MK905406 | Phthiraptera | Pediculidae | Pediculus | humanus capitis | BOLD_all | 384 | Akapala | rudis | 0.596 | Strepsiptera | Stylopidae | Stylops | sp. | 0.9013 |
| MK905406 / RC | Phthiraptera | Pediculidae | Pediculus | humanus capitis | BOLD_all | 384 | | | | Hymenoptera | Eucharitidae | Akapala | rudis | 0.5960 |

RC = reverse complement. Blue and red shading indicate correct or inaccurate identification, respectively, at each taxonomic rank.

## Factors responsible for four 'errors' in generic assignment

Both BOLD and GenBank delivered generic identifications deemed incorrect for four specimens. In each case, the query sequence showed close similarity (95–100% in three cases, 90% in one) to taxa belonging to a different order than that analyzed (Table 1; S1 File and S2 File). These discordances could either reflect errors in the reference libraries or in the query sequences. The cause for one misidentification was certain; it arose through internal cross-contamination as the sequence for *Hexagenia limbata* was a truncated version of that for *Glossina palpalis* (identical at all 250 bp that overlapped). The other three mismatches involved taxa (springtail, gall midge, strepsipteran) unrepresented among the 17 tested species, ruling out contamination between the specimens included in the analyses. Moreover, because of their striking morphological differences to the test taxa (house fly, dragonfly, louse), misidentification can be excluded as a cause. This leaves two possible explanations–contamination in the reference sequence libraries or in the query sequences. Because each query sequence was embedded within many independently generated reference sequences from another order, these cases of misidentification clearly arose from contamination of the query sequences. The contamination of a house fly by collembolan DNA and a dragonfly by gall midge DNA is easily explained by small non-target specimens or their fragments being tangled in the legs of the much larger target specimens. This occurs commonly when specimens are sorted from bulk samples and mounted individually for storage in a natural history collection. In contrast, contamination of *Pediculus humanus* by strepsipteran DNA seems highly unlikely at first glance. However, examination of the loan records at the USNM revealed that the loan of material to Meiklejohn et al. contained multiple specimens which were not included in their analyses or mentioned in their article–and among them, two were Strepsiptera (USNM ENT 01248370 and 01248357). Cross-contamination is a well-recognized risk when working with museum specimens so it is standard practice to check for its occurrence [13,14]. While Meiklejohn et al. [10] exercised some precautions in their laboratory protocols such as the incorporation of negative controls in PCR, there was no evidence that they considered the possibility that some of their DNA sequences derived from non-target taxa. After excluding these four cases, the number of correct identifications for BOLD and GenBank (12/13 for genus, 9/13 for species) was identical.

## Need for taxonomic validation of museum specimens

The four remaining 'incorrect' identifications all involved cases where BOLD and GenBank assigned the query sequence to a species closely related to the taxon analyzed by Meiklejohn et al. [10] (Table 1). As such, the evidence for misidentification rests on the presumption that their specimens were correctly identified. While the National Museum of Natural History is considered one of the better curated of North American insect collections, the quality of the identification of individual specimens depends on the expertise available and the time elapsed since they were assigned to a species [15]. As such, specimens may be misidentified, mirroring the situation reported in other studies. For example, Meier & Dikow [16] found that 12% of all species-level identifications for a genus of asilid flies from various collections were wrong. Similarly, Muona [17] found that from 1–25% of beetles belonging to two easily discriminated species pairs and one species tetrad were incorrectly identified in a major collection. Efforts to build a DNA barcode reference library for North American Lepidoptera exposed many misidentified specimens and overlooked cryptic species in major collections [18]. Importantly, all four cases of apparent misidentification reported by Meiklejohn et al. [10] involve species whose recognition is not straightforward. The sole case of generic misidentification involved a presumptive specimen of the flea, *Ctenocephalides felis*, whose sequence matched those for

another flea, *Pulex irritans*, on BOLD and GenBank. BOLD holds nearly 1,200 records, contributed by 15 institutions, representing four species of *Ctenocephalides* and each possesses a divergent array of barcode sequences. The barcode results support the monophyly of all species in the genus while *P. irritans* forms a sister taxon. The specimen sequenced by Meiklejohn et al. [10] was consumed in the analysis, but we examined the vial of specimens from which it came. The sequence clusters within other independent records of *Pulex irritans*, which was also reported from dogs by the same study [19] from which the specimens in the vial originate. It is likely that a *Pulex* specimen was mis-sorted among the many *Ctenocephalides* in the tube of alcohol. The specimens were collected from Washington County, Arkansas, not from Washington state, as incorrectly noted in the GenBank record.

The three remaining cases of presumptive species-level misidentifications involved genera (*Gryllus*, *Glossina*, *Phanaeus*) with complex taxonomy. One of the three species, *Gryllus assimilis*, was formerly thought to be widely distributed in the New World, but it is now recognized to be a complex of 8+ species, several of which can only be reliably distinguished by their calls or life history [20,21]. Examination of the specimen analyzed by Meiklejohn et al. revealed that it was collected in Virginia–well outside the range of the true *G. assimilis*, which is a more southern species. In the United States, *G. assimilis* is only known from Texas and, as an introduced species, from southern Florida [21]. The identification retrieved for this specimen from both BOLD and GenBank (*G. pennsylvanicus*) is likely correct, although other species do occur in Virginia [21]. Similarly, the query species of tsetse fly (*G. palpalis*) belongs to a complex that includes *G. brevipalpis* [22–24], the species identified as the closest match by BOLD and GenBank. There are hundreds of *Glossina* COI sequences in GenBank and BOLD, but most of them are from the 3' end of the gene and do not overlap with the DNA barcode region. The *Phanaeus* specimen analyzed by Meiklejohn et al. [10] may well be correctly identified as it matches closely (0.9955) to two specimens of *P. vindex* from an earlier study on the phylogeny of this genus [25]. The closest match (0.9982) for this specimen on both BOLD and GenBank is a sequence associated with an interim species name, which results in the apparent species-level misidentification. It should be noted that *Phanaeus vindex* belongs to a group of three closely related species as well as some controversial subspecies [26], but it is likely more diverse as records for it on BOLD include four distinct COI sequence clusters. Because of these taxonomic uncertainties, the four cases of presumptive species- or genus-level misidentifications are best viewed as unconfirmed or incorrect.

## Resolving taxonomic uncertainty

As the preceding section reveals, efforts to assess the resolution of DNA barcodes are often constrained by poor taxonomy. It is certain that some records on BOLD and GenBank derive from misidentified specimens, but there is no easy path to correct them. This fact was powerfully demonstrated by Mutanen et al. [27] in a study of DNA barcode variation in 4,977 species of European Lepidoptera which revealed that 60% of the cases initially thought to indicate compromised species resolution or DNA barcode sharing actually arose as a result of misidentifications, databasing errors, or flawed taxonomy. As the taxonomic system for European Lepidoptera is very advanced, similar issues will be a greater impediment in most other groups. Databases like BOLD and GenBank record these divergences in taxonomic opinion, but they cannot resolve them, providing strong motivation for approaches that sidestep this barrier. The Barcode Index Number (BIN) system is a good candidate as it makes it possible to objectively register genetically diversified lineages [28]. One of the species in the current study, *Forficula auricularia*, provides a good example of the enhanced geographic and taxonomic resolution offered by BINs that could be useful in forensic and many other contexts. This

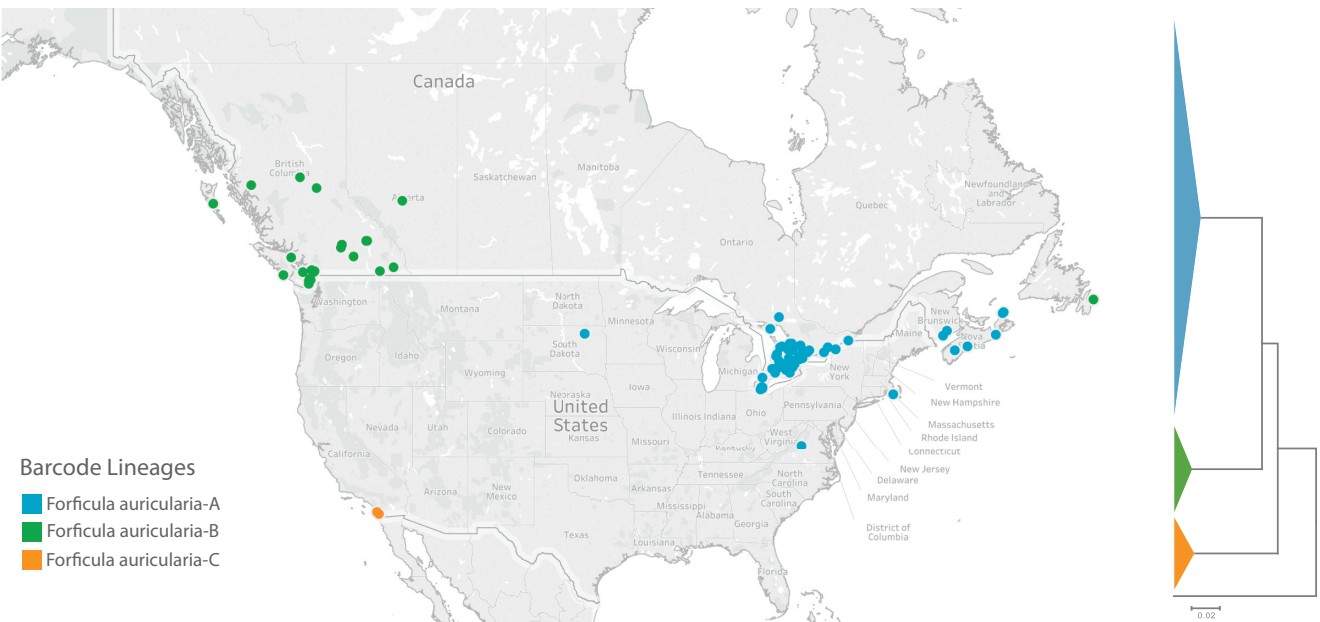

**Fig 1. Geographic distributions and sequence clustering of the three barcode lineages of *Forficula auricularia* in North America.**

taxon was, in principle, correctly identified through GenBank by Meiklejohn et al. [10] (Table 1). However, although only a single species is still formally recognized, *F. auricularia* has been known to include two lineages with differing distributions and life histories for >20 years [29,30]. In fact, barcode results indicate that North American populations actually include three divergent lineages with allopatric distributions (Fig 1). As such, BIN assignments provide information on the geographic distributions of the component lineages of this species complex that could be important in certain contexts, but that would be overlooked by a species-based assignment. Because most species of multicellular organisms await description, it is certain that there are many other cases where BIN-based analysis will enhance geographic resolution.

## Distinction between BOLD and GenBank

It is not surprising that BOLD and GenBank demonstrated similar performance in identification, once operational issues were resolved, as many records appear in both platforms, which are intended to be complementary. Sequences of COI submitted independently to GenBank are mined and entered into BOLD periodically while records from BOLD are submitted to GenBank when they are published. At present, 11% of all COI barcode records on BOLD originate from GenBank, while 75% of the COI barcodes on GenBank derive from BOLD. Although many records are shared, the two platforms diverge in collateral data. For example, for the 17 species of insects analyzed in [10], 65% of the records originating from BOLD possess GPS coordinates, 60% have trace electropherograms, and 40% have specimen images, while only 26% of those originating from GenBank had GPS coordinates and all lacked images and electropherograms. In addition, BOLD employs BINs to integrate records that lack a genus or species designation with those that possess them. These extended data elements and functionality are a valuable, often essential, component in the evaluation of identification results.

## Conclusions and path forward

Six of the 17 species examined by Meiklejohn et al. [10] escaped operational errors, but the other 11 did not (Table 2), explaining the low identification success they reported. Even after correcting for the use of reverse complements, the effectiveness of DNA barcoding could not be evaluated for eight species, those impacted by sequence contamination or taxonomic uncertainty. Importantly, DNA barcode records in BOLD and GenBank did deliver a correct species assignment for the other nine species. While the outcome for these species is reassuring, the lack of an outcome for other taxa reveals the need for improved protocols. Clearly, two conditions need to be satisfied to ensure a correct identification–the query sequences must be legitimate and the reference libraries must be well-validated. As a start, any study that aims to employ DNA barcodes for species identification should include steps to ensure the sequences recovered are valid by including positive and negative controls, by assessing sequence quality, and by checking for contaminants (Fig 2). Presuming the query sequences pass these quality checks, the generation of a reliable identification requires a comprehensive, well-validated reference library. The taxonomic reliability of GenBank has often been questioned [e.g. 31], although the actual overall data quality is much better than often assumed [32]. Because BOLD is a workbench for the DNA barcode research community, it will always contain sequences from specimens whose identifications are being refined. However, the taxonomic coverage and resolution of the COI barcode library on BOLD, and hence the accuracy of identification queries, is steadily improving [33]. The establishment of a Barcode REF library, based upon a small number of carefully validated records for each species, would represent an important step towards further improving BOLD's capacity to generate reliable identifications. Under ideal circumstances, the primary reference sequence for each species would derive from its holotype. However, because 90% of all multicellular organisms await description, and the status of many described species groups is uncertain, these efforts will need to be reinforced by a BIN-based approach.

**Table 2. Three categories of operational errors which compromised efforts by Meiklejohn et al. [10] to test the effectiveness of the BOLD and GenBank reference libraries in identifying 17 insect species.**

| Specimen # | ID | Reverse Complement | Contamination | Incorrect ID |
|---|---|---|---|---|
| 1 | *Phanaeus vindex* | — | — | Yes |
| 2 | *Forficula auricularia* | Yes | — | — |
| 3 | *Chrysomya rufifacies* | — | — | — |
| 4 | *Calliphora vicina* | — | — | — |
| 5 | *Aedes aegypti* | — | — | — |
| 6 | *Glossina palpalis* | — | — | Yes |
| 7 | *Musca domestica* | Yes | Yes | N.D. |
| 8 | *Hexagenia limbata* | — | Yes | N.D. |
| 9 | *Vespula squamosa* | — | — | — |
| 10 | *Callosamia promethea* | — | — | — |
| 11 | *Danaus plexippus* | — | — | — |
| 12 | *Merope tuber* | Yes | — | — |
| 13 | *Ululodes quadripunctatus* | Yes | — | — |
| 14 | *Gomphus exilis* | Yes | Yes | N.D. |
| 15 | *Gryllus assimilis* | — | — | Yes |
| 16 | *Ctenocephalides felis* | Yes | — | Yes |
| 17 | *Pediculus humanus capitis* | Yes | Yes | N.D. |

N.D. = not determined.

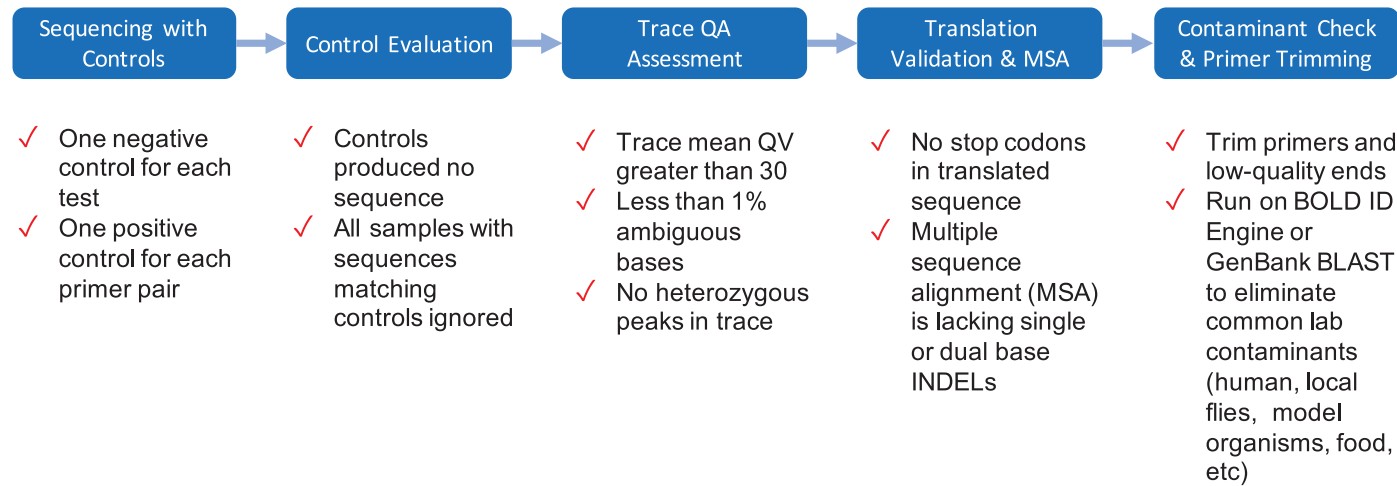

**Fig 2. Five key workflow features to maximize the chance of recovering reliable sequence records.**

## Supporting information

**S1 File. Top 20 matches in GenBank BLAST queries for the four specimens deemed cross-contaminations.**
(XLSX)

**S2 File. Top 20 matches from queries to the BOLD ID engine for four specimens whose COI sequences derive from cross-contamination.**
(XLSX)

## Acknowledgments

Floyd Shockley and Cailin Meyer reassembled the voucher specimens which had been returned to the USNM collection. Nicholas Silverson added images of most of the specimens to BOLD.

## Author Contributions

**Conceptualization:** Paul D. N. Hebert.

**Data curation:** Mikko Pentinsaari.

**Formal analysis:** Mikko Pentinsaari.

**Funding acquisition:** Paul D. N. Hebert.

**Investigation:** Mikko Pentinsaari, Sujeevan Ratnasingham, Scott E. Miller, Paul D. N. Hebert.

**Project administration:** Paul D. N. Hebert.

**Supervision:** Paul D. N. Hebert.

**Visualization:** Sujeevan Ratnasingham.

**Writing – original draft:** Mikko Pentinsaari, Paul D. N. Hebert.

**Writing – review & editing:** Mikko Pentinsaari, Sujeevan Ratnasingham, Scott E. Miller, Paul D. N. Hebert.

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
