## [Decision Letter · Decision Letter 0]

6 Jan 2020

PONE-D-19-20600

Forensics and DNA Barcodes – Do Identification Errors Arise in the Lab or in the Sequence Libraries?

PLOS ONE

Dear Dr. Hebert,

Thank you for submitting your manuscript to PLOS ONE. We invite you to submit a revised version of the manuscript that addresses the points raised during the review process.

Both reviewers are very positive about your work, and I agree that it is important to publish it with minimum delay. Please address in a revised version the minor recommendations from the reviewers.

We would appreciate receiving your revised manuscript by Feb 07 2020 11:59PM. To enhance the reproducibility of your results, we recommend that if applicable you deposit your laboratory protocols in protocols.io, where a protocol can be assigned its own identifier (DOI) such that it can be cited independently in the future. For instructions see: http://journals.plos.org/plosone/s/submission-guidelines#loc-laboratory-protocols

We look forward to receiving your revised manuscript.

Kind regards,

Matjaž Kuntner

Academic Editor

PLOS ONE

Journal Requirements:

Reviewers' comments:

Reviewer's Responses to Questions

**Comments to the Author**

1. Is the manuscript technically sound, and do the data support the conclusions?

Reviewer #1: Yes

Reviewer #2: Yes

2. Has the statistical analysis been performed appropriately and rigorously? 

Reviewer #1: N/A

Reviewer #2: Yes

3. Have the authors made all data underlying the findings in their manuscript fully available?

Reviewer #1: Yes

Reviewer #2: Yes

4. Is the manuscript presented in an intelligible fashion and written in standard English?

Reviewer #1: Yes

Reviewer #2: Yes

5. Review Comments to the Author

Reviewer #1: This is an important paper. It competently and completely addresses the errors and misunderstandings reported by Meiklejohn et al. 2019. The conclusions of the Meiklejohn et al. paper were flawed, full stop, but have the potential to greatly influence the field. Especially given the title. Therefore, a reanalysis of the results is called for and the authors do a thorough job of it. Hopefully their work will be taken into account by future readers of the Meiklejohn paper.

They take the opportunity to provide a pipeline that, if followed, will produce query sequences that are high quality – avoiding contamination, pseudogenes, and ambiguous bases. I expect that the majority of the DNA barcoding community follows a similar pipeline, but it is good to see it laid out so clearly. The pipeline would be a good place to start for a lab just beginning its barcoding efforts.

In the course of the paper, the discussion of the difficulties of using museum specimens as references, especially the pitfalls and effects of mislabeling, are clearly laid out. In conjunction with their discussion of the impact of taxonomic uncertainty, this information lays out the limitations of using DNA barcoding for identification. In general, they make clear the importance of considering these limitations and how they may be project- or taxon- or location-specific. DNA barcoding works very well for many questions, but a researcher needs to be aware.

Reviewer #2: In essence, the submitted manuscript draft of Pentinsaari and coauthors is a rebuttal to a manuscript recently published in PloS One “Meiklejohn KA, Damaso N, Robertson JM (2019) Assessment of BOLD and GenBank – Their accuracy and reliability for the identification of biological materials. PLoS ONE 14(6): e0217084”. In that work, Meiklejohn et al. (2019) assessed the the usability of DNA barcoding coupled with public sequence repositories (GenBank, BOLD) in plants (using several genetic markers rbcL, matK, trnH-psbA, ITS) and insects (using a single genetic marker – COI) for among others the purpose of forensic studies. One of their main conclusion (Meiklejohn et al., 2019) was that the COI coupled with GenBank and BOLD has a very limited power in identification of insect species (53% and 35%, respectively).

To investigate what are the reasons for such a low success Pentinsaari and coauthors reanalyzed the COI data set from Meiklejohn et al. (2019) and identified two grave errors in their data processing that lead to such a a low success of COI DNA barcoding. The first error was that in their research Meiklejohn et al. (2019) obviously submitted a reverse complement rather than the coding sequence into the BOLD engine (which raised the success of species/genus level identification for both approaches to over 50/70%) demonstrated. Furthermore, they that additional four misidentifications (23%) were due cross-contamination of the museum samples used for DNA extractions. The remaining species level misidentifications were due misidentification of the museum specimens or a result of complex not fully resolved species level taxonomies. In addition, the authors give a suggestions on how can forensic sciences make of DNA Barcoding with the help of Barcode Index Number (BIN) system even when analyzing species with taxonomies that are not fully resolved. Finally, the authors give a workflow with a series of recommendations to maximize the chance of recovering reliable sequence records for DNA barcoding.

I have carefully read both the Meiklejohn et al. (2019) paper and the draft Pentinsaari and coauthors have submitted. Furthermore, to leave any doubt I have also personally rechecked and reanalyzed the COI data set in question and can confirm their results and I also agree with their identifications of errors in Meiklejohn et al. (2019). Therefore I, recommend the editor to accept this manuscript after a minor revision - I am adding a couple of suggestions that would make it a bit easier to follow the what they have done

Here are some minor recommendations

Results and Discussion

The authors might also want to mention that the data from Meiklejohn et al. (2019) is not deposited in BOLD and thus trace files are also not available (at least I could not find them). Namely, electopherograms could also be used to see if there are double peaks indicating chimeric sequences due to such contamination.

155 I would avoid using ‘species’ (I am reffering to ‘’). Even if we know that this taxon name probably referees to several related it is still the only one described. Also such situations are not rare and ‘’ doesn’t really clarify what the authors think is wrong with it. Maybe you can simply say “… Forficula auricularia complex, which was also used in the current study provides a good example...”

155-157: Maybe this is not the best example as Forficula auricularia was also correctly identified by Meiklejohn et al. (2019). But, yes it is a good example for BIN system. I suggest the authors recognize that Meiklejohn identified this species correctly somewhere in the text.

Supplementary Table 1

I suggest to include the Table S1 into the main text as it is integral to the understanding of the manuscript and readers. Furthermore, it will be easier to follow manuscript if the authors refer to this table when discussing different categories of mistakes through the text (reverse complement, likely contamination, misidentification).

Figure 2:

One negative control for each test same – what is the meaning of “same”.

No heterozygous peaks in trace – in mtDNA there are (almost never) no heterzoygous peaks. Such peaks are due to pseudogenes or contamination. Here they should be referred to as chimeric or by using another incorporate term.

6. PLOS authors have the option to publish the peer review history of their article (what does this mean?). If published, this will include your full peer review and any attached files.

Reviewer #1: No

Reviewer #2: No

---

## [Author Response · Author response to Decision Letter 0]

12 Jan 2020

To the editor:

We thank the reviewers for scrutinizing our manuscript and providing constructive feedback. We agree with the suggestions made by Reviewer 2, and have edited the manuscript text and Figure 2 accordingly. We have also revised the formatting of the manuscript file to comply with the PLOS style requirements as requested by the editor. All changes are documented in the annotated version of the revised manuscript file.

Sincerely,

Mikko Pentinsaari

Sujeevan Ratnasingham

Scott E. Miller

Paul D. N. Hebert

---

## [Decision Letter · Decision Letter 1]

10 Feb 2020

PONE-D-19-20600R1

Forensics and DNA barcodes – do identification errors arise in the lab or in the sequence libraries?

PLOS ONE

Dear Dr. Hebert,

Thank you for making the corrections the the originally submitted version of your manuscript. Before we move towards acceptance of your paper, please consider the following. As per PLoS policy, the journal editors have asked the authors of the paper you critique to provide a signed review. This review is now available to you herein. Please take some time to provide a list of rebuttals to this review, if you feel so, or changes to your manuscript, where appropriate and where you agree with these suggestions.

We would appreciate receiving your revised manuscript by Mar 26 2020 11:59PM. To enhance the reproducibility of your results, we recommend that if applicable you deposit your laboratory protocols in protocols.io, where a protocol can be assigned its own identifier (DOI) such that it can be cited independently in the future. For instructions see: http://journals.plos.org/plosone/s/submission-guidelines#loc-laboratory-protocols

We look forward to receiving your revised manuscript.

Kind regards,

Matjaž Kuntner

Academic Editor

PLOS ONE

Reviewers' comments:

Reviewer's Responses to Questions

**Comments to the Author**

1. If the authors have adequately addressed your comments raised in a previous round of review and you feel that this manuscript is now acceptable for publication, you may indicate that here to bypass the “Comments to the Author” section, enter your conflict of interest statement in the “Confidential to Editor” section, and submit your "Accept" recommendation.

Reviewer #3: (No Response)

2. Is the manuscript technically sound, and do the data support the conclusions?

Reviewer #3: Partly

3. Has the statistical analysis been performed appropriately and rigorously? 

Reviewer #3: N/A

4. Have the authors made all data underlying the findings in their manuscript fully available?

Reviewer #3: Yes

5. Is the manuscript presented in an intelligible fashion and written in standard English?

Reviewer #3: Yes

6. Review Comments to the Author

Reviewer #3: ¬¬Review of: Forensics and DNA barcodes – do identification errors arise in the lab or in the sequence libraries?

PONE-D-19-20600R1

Paper summary:

This paper seeks to identify reasons why the results presented in Meiklejohn et al (PLoS One. 2019;14: e0217084. Doi:10.1371/journal.pone.0217084) provide a misleading evaluation of the data contained in BOLD. In Meiklejohn et al, appropriate DNA barcode sequences from 17 insect, 61 plant and 16 macro-fungi species (curated reference material sourced from national collections) were obtained and compared to both GenBank and BOLD. Specifically for insect taxa, Meiklejohn et al reported that GenBank out performed BOLD with respect to correct species level identifications. In the paper under review, the authors identify three main reasons for the misidentifications reported in Meiklejohn et al using BOLD: 1) reverse complement sequences were searched against BOLD, 2) sequences were cross-contaminated, and 3) taxa included were either misidentified or part of a species complex.

Major Technical Comments:

1) Line 24-26: The authors made the claim that they have accounted for all of the misidentifications in Meiklejohn et al. by identifying errors in the methods (i.e., “problems in the reference libraries or in the query sequences used to test them”). This idea is later contradicted in line 48 when they restated this as “..most, if not all, of the misidentifications” and later in line 82-83 where the authors implied that misidentifications also occurred due to “errors in the reference libraries..”, which was one of the primary conclusions of Meiklejohn et al. The authors contradicted themselves throughout their manuscript and need to update the text for consistency (based on valid data and conclusions – discussed in later points).

2) Line 41-42: Authors provided the statement, “Because COI generally has high accuracy in species assignment [4–9], the conclusions from a recent study by Meiklejohn et al. [10] were surprising.”. This statement would lead readers to believe that Meiklejohn et al did not acknowledge that their results did not conform to the majority of previous studies examining the accuracy of BOLD. This is not the case, as Meiklejohn et al. highlighted this by stating “This result was lower than previously reported for flies (Diptera [15,20], beetles (Coleoptera [23]), butterflies and moths (Lepidoptera [24]), when searching against either or both of these databases.” Thus, the text needs to be modified to reflect that Meiklejohn et al. also reported that their results were lower than previously reported.

3) Line 71-75: It is apparent that Meiklejohn et al did indeed search the reverse complement for seven COI sequences against BOLD. In doing so, the matches they returned were lower with respect to the similarity statistic and were to the incorrect species (highlighted by the authors in Table 1). While it is good scientific practice to search sequences in the forward orientation, at the time in which Meiklejohn et al searched sequences against BOLD (reported in their paper as “July 2017-January 2018”), there was no guidance or requirement that COI sequences must be searched in the forward orientation; neither Ratnasingham and Hebert (2007) nor the “BOLD Print Handbook for BOLD (v3 or v4)” specified the orientation of query sequences. Further, it is apparent the identification interface for COI on boldsystems.org was updated only ~6 months ago (after Meiklejohn et al was published) to state “Enter fasta formatted sequences in the forward orientation” and if a reverse complement sequence is submitted the result states “These results may be due to reverse or reverse complement sequences. Please try again with the forward orientation”. Conversely, the BOLD database always accommodated both orientations for plants and fungal identification engines. Thus, when researchers completed the searches against the original BOLD, they would have had no reason to immediately assume that the incorrect matches were a result of query sequence orientation. While the authors are correct in identifying a source of error in Meiklejohn et al, they have neglected to address why this mistake might have occurred. Given it was not previously explicitly stated that forward sequences must be searched (solely for insect COI sequences; not fungal or plant sequences), it is possible that results from other researchers might also have been affected by this issue and so must be addressed in this manuscript.

Meiklejohn et al. wrote that the BOLD similarity metric was an indicator of an incorrect match and recommended that it is important to use the metric to determine confidence of the results. In Meiklejohn et al., results that gave lower similarity were not excluded, presumably in order to see what the database results gave as the “most similar match”.

4) Line 89-91: The authors made a bold statement and suggested that “misidentification clearly arose from contamination” and made a reasonable hypothesis as to a potential cause for misidentifications. However, they failed to meet their burden of proof as well as contradicted themselves later in the article. In line 85-87, the authors admitted that “the other three mismatches involve taxa unrepresented among the 17 tested species ruling out internal contamination.” If internal contamination were the cause of one sample’s misidentification, then the authors need to provide an explanation for the lack of this contamination in the other samples. It is at this point that the authors contradicted themselves and by writing, “This leaves two possible explanations-contamination in the reference sequences libraries or in the query sequences.” Again, the authors have not met their burden of proof for either claim of contamination. Furthermore, misidentification caused by inconsistencies in reference sequence libraries was a major conclusion cited by Meiklejohn et al. as a source for misidentification in the first place. Furthermore, Meiklejohn et al. used controls during sample processing, a procedure that was referenced in their material and methods (see further explanation below). Even after making these concessions, the authors in line 91 wrote, “these cases of misidentification clearly arose from contamination of query sequences,” again, contradicting themselves.

For the four specimens in which the COI sequences were identified by the authors as stemming from cross-contamination, only one of those sequences corresponded to a species processed in Meiklejohn et al (Hexagenia limbata). Typically, inadvertent cross-contamination arises when a laboratory routinely processes large numbers of similar samples. Do the authors know whether the research division of the FBI Laboratory (where Meiklejohn et al performed the research) processes large numbers of insect material? If so, they should state that as a plausible source of DNA for cross-contamination, rather than just providing a broad sweeping explanation of “cross-contamination”.

Overall for this section (subheading “Factors responsible for four ‘errors’ in generic assignment”) the authors need to make clear and well-supported arguments, rather than swapping between possible explanations for the misidentifications and contradicting themselves. Substantial modifications to the text are needed to address this.

5) Line 91-94: Authors wrote, “Cross-contamination is a well- recognized risk when working with museum specimens, so it is standard practice to check for its occurrence [13,14], but Meiklejohn et al. [10] make no mention of exercising precautions in this regard.” Actually, Meiklejohn et al. did exercise some of the precautions identified in Figure 2, but these might have been overlooked by the authors of this paper as Meiklejohn et al referenced their original paper for the majority of their methods (Meiklejohn et al 2018, International Journal of Legal Medicine). Perhaps Meiklejohn et al should have rearticulated these precautions in their 2019 paper, but they did include some such as negative controls, bleach specimen wash prior to extraction, and amplification using various primer pairs. The authors need to modify the text either to state which precautions were taken by Meiklejohn et al, or remove the text ‘make no mention of exercising precautions in this regard’.

6) Line 98-103: Authors briefly discussed how even at a well curated museum there is a potential for taxonomic misidentification, particularly with older samples (see Line 102, “As such, specimens may be misidentified, mirroring the situation reported in other studies”). This reviewer acknowledges that this can be true, however it would be unreasonable for Meiklejohn et al. to second-guess the taxonomic classification given to a specimen at a museum by a subject matter expert. To confirm the authors’ hypothesis, they could have rechecked the identity of the vouchered specimen. Since it is standard practice to return unused specimen material to the repository, it would have been straight-forward for the authors to confirm their claims of misidentification by examination of the returned specimens. Since the authors did not report doing this examination, their assumptions are overreaching.

7) Line 108: Authors wrote, “Importantly, all four cases of apparent misidentification reported by Meiklejohn et al. [10] involve species whose recognition is not straightforward.” Upon review, Meiklejohn et al reported that common insect species were selected for inclusion and discussed that possible misidentifications (i.e., sister species) might have impacted identification success. However, if the species were incorrectly identified as the sister species, correct genus level identification still would have been obtained (this was not the case for four taxa in question). Thus, the authors need to update the text to accurately reflect that the type of species included in Meiklejohn et al were common ones.

8) Line 118: Authors state “the supposed specimen..analyzed by Meiklejohn et al is almost certainly P. irritans”. This reviewer does not know how the authors could possibly make this bold claim without having access to the specimen used in Meiklejohn et al to confirm the identity morphologically. Moreover, as stated in the above comment, it would be inappropriate to second guess the identification as outlined above. In retrospect, this supports the claim made by Meiklejohn et al. that care must be taken when using barcoding as the sole source for identifying material found in the environment, due to a wide variety of sources for possible misidentifications including morphological misidentification in barcode and genetic databases.

9) Line 125-127: Authors argued that these specific instances should be viewed as “unconfirmed”. However, these authors are leading entomologists. One could argue how would non-entomologists (such as Meiklejohn et al.) differentiate between an unconfirmed and a misidentification? Authors themselves mention in Line 36-39 that DNA based approaches to species identification can help fill the lack of taxonomic expertise that often impedes analysis. It is apparent from reviewing the manuscript that the purpose of the Meiklejohn et al. paper was different, as they were focused on the general accuracy of GenBank and BOLD for returning a match to the expected species, not providing guidance on how matches should be reported out. Thus, the authors need to either remove the term ‘unconfirmed’ or explain how a user of bold with no entomology expertise could differentiate between an ‘unconfirmed’ and a ‘misidentification’.

10) Line 128-149: This reviewer agrees that identification of biological materials for forensics should be tackled interdisciplinary and with common sense, by looking at geographic information, etc. However, it is incorrect to suggest that Meiklejohn et al. scope was discussing forensics and DNA barcodes as their paper was solely on the utility of the DNA barcoding databases using a subset of curated museum species. It is unjust to assume the co-authors of Meiklejohn et al wrote in terms of forensics because of their affiliation. Presumably, Meiklejohn et al. would have been more specific regarding cautions if their objective was for forensic application. However, this was not the case. The authors need to modify the text to reflect the inaccuracy.

11) Line 160-166: Authors briefly summarized how it is possible to use GPS coordinate metadata included in BOLD to better source and identify samples. This is a reasonable suggestion for those who wish to more accurately identify a species for purely taxonomic reasons. However, doing so would require subject expertise in entomology or arthropod evolution to properly utilize this type of data (e.g., specimen images and GPS coordinates) which is not reasonable to expect for most users of BOLD whose background knowledge will vary wildly from one user to the next. Finally, this kind of analysis was outside the scope of work performed by Meiklejohn et al. and therefore an unreasonable criticism of their methodology. This section either needs to be removed, or placed within context of the goal/purpose of the Meiklejohn et al study.

12) Line 168-171: This statement overstates that 11 species did not escape operational errors. The sequences for only 3 species were able to be fully resolved, but the sequences for the other 8 species were not. Authors failed to meet burden of proof as well as contradicted themselves throughout the paper writing that these species were “not as straightforward” and should be viewed as “unconfirmed”.

Minor Technical Comments:

1) Line 55-59: The impression the authors made here is that Meiklejohn et al used old specimens and thus this contributed to their misidentification results. Meiklejohn et al. included the age of the specimens as metadata and highlighted that amplicon/sequence length recovery was not correlated with specimen age. Thus, the authors need to modify the text to include this fact.

2) Line 80: Awkward wording of sentence. “Both BOLD and GenBank delivered generic identifications deemed incorrect for four specimens.” Not sure what this sentence means.

3) Line 155-158: Authors explain how BOLD and GenBank are similar. This is also mentioned in Meiklejohn et al and could have ben acknowledged.

4) Line 175-178: Authors suggest that Meiklejohn et al. did not use proper controls and quality checks; however, they did this according to their protocols that was cited in Methods. Meiklejohn et al out-referenced these steps from their earlier paper (Meiklejohn et al 2018 International Journal of Legal Medicine). The authors need to update the text to reflect this.

Comments on Tables and Figures:

Table 1: When checking the matches returned from the sequences listed in Table 1, it is apparent the authors mislabeled the query sequences. For example, upon downloading MK905402 from GenBank and searching against BOLD one returns a match for Forifucla auricularia. Thus, the authors need to place the red results from Meiklejohn et al under the MK905402/RC row for BOLD_all, not the MK905402 row alone.

Lastly, this table shows that 7 of 11 needed to be RC. However, only 3 species were able to be resolved. All 7 sequences that were not RC (as Meiklejohn et al. stated in their Discussion) were able to be confidently known to be incorrect due to their similarity statistic (<60%). Furthermore, the sequence orientation for BOLD was not explicated stated at the time Meiklejohn et al. queried the sequences and seems to only be affecting insect sequences (the BOLD fungal and plant algorithm was not sensitive to sequence orientation).

Table 2: What is N.D? the table description states “three categories of operational errors which compromised efforts by Meiklejohn et al…” Given this reviewer has concerns with the claims (and contradictions) the authors made about contamination, the ‘contamination’ column in this table needs to be removed. For instance, in line 85-87, the authors admitted that “the other three mismatches.. along the 17 tested species ruling out internal contamination.” Therefore, these species should not be included in the list as “contaminated” in the table.

Moreover, it is unclear what comprises the third operational error. Is it incorrect identification? Incorrect identification is not due to Meiklejohn et al. as these authors also had the same results. Neither Meiklejohn et al nor the authors performed taxonomic identification of the specimens. As presented, it appears to this reviewer that the authors wanted to find any reason for misidentification, aside from possible issues with BOLD. The column title needs to be modified accordingly.

Figure 2: This is a great figure to show proper workflow. Meiklejohn et al. followed many of these features (e.g., negative controls, quality assessment, etc.). Also given their sequences were submitted to GenBank and are from a coding region, the sequences would also have been checked for premature stop codons, indels, etc. (this is part of GenBank’s QC process before acceptance).

Additional Comments:

Appropriateness of the Methods

There has been a shift in the guidelines for using BOLD for COI identifications – the interface now states that sequences must be submitted in forward orientation. The authors need to highlight that they searched the BOLD when this requirement had been articulated.

To support the claims made in this paper specifically on misidentification and cross contamination, the authors should have gone back to the pinned vouchered material at USNM to a) taxonomically confirm the species (addressing misidentification errors), and b) independently resequencing the questionable specimens (addressing cross-contamination issues). The addition of these steps would provide the authors evidence to classify the errors reported by Meiklejohn et al. A description of this work was not given.

Support of the Conclusions by the Results

The authors identified three reasons for the misidentification reported in Meiklejohn et al: 1) reverse complement sequences were searched against BOLD, 2) sequences were cross-contaminated, and 3) taxa included were either misidentified or part of a species complex. The authors results supported conclusion 1), however only circumstantial evidence/results were provided to support conclusions 2 and 3.

Over reaching Conclusions

Several conclusions appear overreaching, especially those on misidentifications and cross-contamination. To validate these conclusions as previously stated, the authors should have gone back to the pinned vouchered material at USNM repository to a) taxonomically confirm the species (validating whether the specimens initially tested were indeed misidentified), and b) independently resequencing from questionable specimens (addressing cross-contamination issues).

7. PLOS authors have the option to publish the peer review history of their article (what does this mean?). If published, this will include your full peer review and any attached files.

Reviewer #3: Yes: James M Robertson

---

## [Author Response · Author response to Decision Letter 1]

26 Mar 2020

1) Line 24-26: The authors made the claim that they have accounted for all of the misidentifications in Meiklejohn et al. by identifying errors in the methods (i.e., “problems in the reference libraries or in the query sequences used to test them”). This idea is later contradicted in line 48 when they restated this as “…most, if not all, of the misidentifications” and later in line 82-83 where the authors implied that misidentifications also occurred due to “errors in the reference libraries...”, which was one of the primary conclusions of Meiklejohn et al. The authors contradicted themselves throughout their manuscript and need to update the text for consistency (based on valid data and conclusions – discussed in later points).

Pentinsaari et al: Public databases will always contain misidentified or otherwise problematic sequence records. Despite this fact, all incorrect identifications noted by Meiklejohn et al. can still be confidently attributed to analytical errors or questionable identifications of their voucher specimens, as detailed in the manuscript and in our replies to the reviewer’s comments below. The sentence in the abstract has been edited to “…responsible for most misidentifications…” since the original wording could be interpreted as not covering misidentifications of the voucher specimens themselves. 

2) Line 41-42: Authors provided the statement, “Because COI generally has high accuracy in species assignment [4–9], the conclusions from a recent study by Meiklejohn et al. [10] were surprising.”. This statement would lead readers to believe that Meiklejohn et al did not acknowledge that their results did not conform to the majority of previous studies examining the accuracy of BOLD. This is not the case, as Meiklejohn et al. highlighted this by stating “This result was lower than previously reported for flies (Diptera [15,20], beetles (Coleoptera [23]), butterflies and moths (Lepidoptera [24]), when searching against either or both of these databases.” Thus, the text needs to be modified to reflect that Meiklejohn et al. also reported that their results were lower than previously reported. 

Pentinsaari et al: A note acknowledging the statement by Meiklejohn et al. has been added near to the end of the introduction.

3) Line 71-75: It is apparent that Meiklejohn et al did indeed search the reverse complement for seven COI sequences against BOLD. In doing so, the matches they returned were lower with respect to the similarity statistic and were to the incorrect species (highlighted by the authors in Table 1). While it is good scientific practice to search sequences in the forward orientation, at the time in which Meiklejohn et al searched sequences against BOLD (reported in their paper as “July 2017-January 2018”), there was no guidance or requirement that COI sequences must be searched in the forward orientation; neither Ratnasingham and Hebert (2007) nor the “BOLD Print Handbook for BOLD (v3 or v4)” specified the orientation of query sequences. Further, it is apparent the identification interface for COI on boldsystems.org was updated only ~6 months ago (after Meiklejohn et al was published) to state “Enter fasta formatted sequences in the forward orientation” and if a reverse complement sequence is submitted the result states “These results may be due to reverse or reverse complement sequences. Please try again with the forward orientation”. Conversely, the BOLD database always accommodated both orientations for plants and fungal identification engines. Thus, when researchers completed the searches against the original BOLD, they would have had no reason to immediately assume that the incorrect matches were a result of query sequence orientation. While the authors are correct in identifying a source of error in Meiklejohn et al, they have neglected to address why this mistake might have occurred. Given it was not previously explicitly stated that forward sequences must be searched (solely for insect COI sequences; not fungal or plant sequences), it is possible that results from other researchers might also have been affected by this issue and so must be addressed in this manuscript.

Meiklejohn et al. wrote that the BOLD similarity metric was an indicator of an incorrect match and recommended that it is important to use the metric to determine confidence of the results. In Meiklejohn et al., results that gave lower similarity were not excluded, presumably in order to see what the database results gave as the “most similar match”. 

Pentinsaari et al: It is common practice that sequence data are analyzed in the forward orientation unless there is a specific reason to do otherwise. The BOLD user interface was updated during the preparation of our manuscript specifically to reduce the likelihood of further occurrences of similar analytical errors. A note on this update has been added in the revised version of our manuscript. We emphasize that BOLD has been employed as a basis for more than 2000 publications since 2007 and we have not encountered any other case where results based on reverse complement sequences were published. 

Although BOLD did not specifically include a warning about the possibility of introducing errors if sequences were injected in the wrong orientation at the time of the original analysis, we still find it surprising that Meiklejohn et al. did not question their results and notice the errors in sequence orientation. Their specimens were expected to match closely to existing data on BOLD, but their closest matches obtained for these incorrect queries were very distant in both taxonomy and sequence similarity. For example, the closest match for the reverse complement sequence of the cat flea (Ctenocephalides felis) specimen retrieved through the BOLD ID engine was the snake Natrix tessellata (similarity 0.6176). The GenBank BLAST query of the same sequence by Meiklejohn et al. identified it as Pulex irritans (0.9642), which at least represents the same family as C. felis. The reasons for this particular discordance in genus/species identification are addressed elsewhere in our manuscript and in this document. GenBank automatically accounts for incorrect orientation of the query sequences which is why the observed results of the BLAST queries for these specimens conformed to expectations better than the BOLD ID engine results (although three of the reverse complement sequences still represented non-target organisms). 

4) Line 89-91: The authors made a bold statement and suggested that “misidentification clearly arose from contamination” and made a reasonable hypothesis as to a potential cause for misidentifications. However, they failed to meet their burden of proof as well as contradicted themselves later in the article. In line 85-87, the authors admitted that “the other three mismatches involve taxa unrepresented among the 17 tested species ruling out internal contamination.” If internal contamination were the cause of one sample’s misidentification, then the authors need to provide an explanation for the lack of this contamination in the other samples. It is at this point that the authors contradicted themselves and by writing, “This leaves two possible explanations-contamination in the reference sequences libraries or in the query sequences.” Again, the authors have not met their burden of proof for either claim of contamination. Furthermore, misidentification caused by inconsistencies in reference sequence libraries was a major conclusion cited by Meiklejohn et al. as a source for misidentification in the first place. Furthermore, Meiklejohn et al. used controls during sample processing, a procedure that was referenced in their material and methods (see further explanation below). Even after making these concessions, the authors in line 91 wrote, “these cases of misidentification clearly arose from contamination of query sequences,” again, contradicting themselves.

For the four specimens in which the COI sequences were identified by the authors as stemming from cross-contamination, only one of those sequences corresponded to a species processed in Meiklejohn et al (Hexagenia limbata). Typically, inadvertent cross-contamination arises when a laboratory routinely processes large numbers of similar samples. Do the authors know whether the research division of the FBI Laboratory (where Meiklejohn et al performed the research) processes large numbers of insect material? If so, they should state that as a plausible source of DNA for cross-contamination, rather than just providing a broad sweeping explanation of “cross-contamination”. 

Overall for this section (subheading “Factors responsible for four ‘errors’ in generic assignment”) the authors need to make clear and well-supported arguments, rather than swapping between possible explanations for the misidentifications and contradicting themselves. Substantial modifications to the text are needed to address this. 

Pentinsaari et al: We state in the beginning of this section that there are two possible explanations for the incorrect genus-level identification for these four sequences. We continue by evaluating which of these two potential error sources is more likely responsible for the observed misidentifications. As we conclude that cross-contamination is more likely in these cases than gross misidentification of the analyzed specimens, we see no signs of the self-contradiction the reviewer suggests.

On line 90, we erroneously stated that the flea sequence was one of the cases affected by cross-contamination. We were, in fact, referring to the Pediculus humanus specimen and should have written “louse” instead of “flea”. This error has been corrected in the revised version of the manuscript. 

In bulk samples of arthropods, small and slender specimens (or fragments of such specimens) often get tangled in the legs or other structures of larger specimens, and will remain attached to the larger specimens when they are mounted for permanent storage in a natural history collection. This is the most likely source of the non-target DNA for Musca domestica and Gomphus exilis. The query sequence for Musca domestica matches the family Isotomidae (Collembola), and the query sequence for Gomphus exilis matches Cecidomyiidae (Diptera). The size difference between target and non-target is an order of magnitude in both cases. Examination of the voucher specimens at USNM revealed that the Gomphus exilis specimen was collected with a Malaise trap in which Cecidomyiidae are often very abundant (see e.g. Hebert et al. 2016, doi: 10.1098/rstb.2015.0333). The collecting method for the Musca domestica specimen was not reported on its label. Fragments of the smaller contaminant specimens were probably attached to a leg of the larger specimens used for DNA extraction, and the contaminant DNA was amplified in PCR instead of that from the target specimens due to differences in primer binding or DNA quality. By examining loan records at the National Museum of Natural History, we found a likely explanation for otherwise unexplainable contamination of Pediculus humanus by strepsipteran DNA. Specifically, we found that two specimens of Strepsiptera were included in the original loan to Meiklejohn et al. Although these specimens were not reported as analyzed in the study and are not mentioned in the article by Meiklejohn et al, it seems likely that a data entry error or lab mishap explains the attribution of a strepsipteran sequence to a louse.

Although the exact source of contamination cannot be proven, it is certain these sequences did not derive from their supposed source organism as they cluster within large numbers of independently generated sequences from a completely different taxon, and are distant from other representatives of the target taxon (again, multiple independently sequenced specimens). Further notes on the possible sources of non-target DNA have been added in our revised manuscript.

5) Line 91-94: Authors wrote, “Cross-contamination is a well- recognized risk when working with museum specimens, so it is standard practice to check for its occurrence [13,14], but Meiklejohn et al. [10] make no mention of exercising precautions in this regard.” Actually, Meiklejohn et al. did exercise some of the precautions identified in Figure 2, but these might have been overlooked by the authors of this paper as Meiklejohn et al referenced their original paper for the majority of their methods (Meiklejohn et al 2018, International Journal of Legal Medicine). Perhaps Meiklejohn et al should have rearticulated these precautions in their 2019 paper, but they did include some such as negative controls, bleach specimen wash prior to extraction, and amplification using various primer pairs. The authors need to modify the text either to state which precautions were taken by Meiklejohn et al, or remove the text ‘make no mention of exercising precautions in this regard’.

Pentinsaari et al: Our text has been modified to mention the use of desirable precautions such as negative controls. However, none of the stated precautions will achieve reliable results if the tissue sample originates from the wrong specimen, multiple specimens, or a misidentified specimen. 

6) Line 98-103: Authors briefly discussed how even at a well curated museum there is a potential for taxonomic misidentification, particularly with older samples (see Line 102, “As such, specimens may be misidentified, mirroring the situation reported in other studies”). This reviewer acknowledges that this can be true, however it would be unreasonable for Meiklejohn et al. to second-guess the taxonomic classification given to a specimen at a museum by a subject matter expert. To confirm the authors’ hypothesis, they could have rechecked the identity of the vouchered specimen. Since it is standard practice to return unused specimen material to the repository, it would have been straight-forward for the authors to confirm their claims of misidentification by examination of the returned specimens. Since the authors did not report doing this examination, their assumptions are overreaching.

7) Line 108: Authors wrote, “Importantly, all four cases of apparent misidentification reported by Meiklejohn et al. [10] involve species whose recognition is not straightforward.” Upon review, Meiklejohn et al reported that common insect species were selected for inclusion and discussed that possible misidentifications (i.e., sister species) might have impacted identification success. However, if the species were incorrectly identified as the sister species, correct genus level identification still would have been obtained (this was not the case for four taxa in question). Thus, the authors need to update the text to accurately reflect that the type of species included in Meiklejohn et al were common ones.

Pentinsaari et al: No taxonomy-related data apart from the species names are reported by Meiklejohn et al. for any of the analyzed specimens, which makes it impossible to evaluate the validity of the identifications based on the information provided in the original study. If the analyzed specimens have been authoritatively identified, the name of the identifier and date (or at least year) of identification would normally be reported on a “det label” together with the taxon name. Based on examination of the label data of the specimens at USNM, this information is available for some, but not all, of the specimens. 

As for provenance data, only the year of collection was reported by Meiklejohn et al, but not the geographic origin of the specimens. The USNM specimen ID numbers of the analyzed specimens were also not listed by Meiklejohn et al. This made it difficult to confirm the source specimen for certain sequences because multiple specimens of some species were included in the loan to Meiklejohn et al., but only a single specimen per species was reported in their analyses, itself a weak approach. In some cases (e.g. louse, flea), USNM loan documentation indicates that the specimens were analyzed destructively so it was not possible to verify their identity morphologically (but see our replies to points 4 and 8). We have mined the sequence records from the original study from GenBank and compiled them on BOLD as a dataset (DS-FBI2019). When possible, specimen metadata have been added to the BOLD records based on the USNM specimen labels.

As detailed in our manuscript, some of the analyzed species are known to belong to complexes of several closely related species where morphological identification is challenging and attempting to only sample supposedly common species does not help avoid this problem.

The Phanaeus vindex specimen analyzed by Meiklejohn et al. clusters in the same BIN with specimens carrying one interim name in addition to P. vindex. The specimen included in the Meiklejohn et al. study happens to be most similar to the specimen with an interim name, hence the observed ‘misidentification’. The Phanaeus vindex species group contains three closely related species currently considered valid, but also several synonymized species names and controversial subspecies (Edmonds & Zídek 2012, http://digitalcommons.unl.edu/insectamundi/784). Other BIN clusters on BOLD also currently carry the name P. vindex, indicating that there likely are more species within this group than is currently recognized. Our revised manuscript includes further details on this species group.

In the case of Gryllus, reliable identifications cannot be based on morphology alone as male song patterns are crucial for confident species identifications without molecular analysis. Several species are now recognized within what was once thought to be a single widespread and common species called G. assimilis. However, based on the specimen label data, the analyzed Gryllus “assimilis” specimen was collected in Virginia, which is well outside the range of the true G. assimilis according to the recent revision of the genus (Weissman & Gray 2019, doi: 10.11646/zootaxa.4705.1.1). The identification retrieved for this specimen from both BOLD and GenBank (Gryllus pennsylvanicus) is likely correct, although other species also occur in eastern United States.

8) Line 118: Authors state “the supposed specimen.analyzed by Meiklejohn et al is almost certainly P. irritans”. This reviewer does not know how the authors could possibly make this bold claim without having access to the specimen used in Meiklejohn et al to confirm the identity morphologically. Moreover, as stated in the above comment, it would be inappropriate to second guess the identification as outlined above. In retrospect, this supports the claim made by Meiklejohn et al. that care must be taken when using barcoding as the sole source for identifying material found in the environment, due to a wide variety of sources for possible misidentifications including morphological misidentification in barcode and genetic databases.

Pentinsaari et al: As the flea specimen was completely consumed in the analysis by Meiklejohn et al, its identification cannot be validated morphologically. Our conclusion that the specimen was likely misidentified is based on comparison of the query sequence to multiple independently generated sequences of both Ctenocephalides and Pulex in both BOLD and GenBank. The query sequence of the supposed C. felis specimen is deeply divergent from all other representatives of this species and its congeners, and instead clusters within sequences identified as P. irritans. Misidentification of the single specimen studied by Meiklejohn et al. is a far more likely cause for this conflict than the alternative explanation that all other records in the databases originating from multiple independent studies are consistently and concordantly misidentified. Although the specimen sequenced by Meiklejohn et al. was consumed, we examined the vial of specimens at USNM from which it came. Pulex irritans was also reported from dogs in the same study (Schiefer & Lancaster 1970, Journal of the Kansas Entomological Society 43:177-181). It is likely that a Pulex specimen was mis-sorted among the many Ctenocephalides in the tube of alcohol. 

9) Line 125-127: Authors argued that these specific instances should be viewed as “unconfirmed”. However, these authors are leading entomologists. One could argue how would non-entomologists (such as Meiklejohn et al.) differentiate between an unconfirmed and a misidentification? Authors themselves mention in Line 36-39 that DNA based approaches to species identification can help fill the lack of taxonomic expertise that often impedes analysis. It is apparent from reviewing the manuscript that the purpose of the Meiklejohn et al. paper was different, as they were focussed on the general accuracy of GenBank and BOLD for returning a match to the expected species, not providing guidance on how matches should be reported out. Thus, the authors need to either remove the term ‘unconfirmed’ or explain how a user of bold with no entomology expertise could differentiate between an ‘unconfirmed’ and a ‘misidentification’.

Pentinsaari et al: By ‘unconfirmed’ we mean an unconfirmed case of DNA barcodes failing to distinguish between species. The goal of the original study as stated by the authors was to assess the accuracy of BOLD and GenBank, not to identify unknown specimens. The accurate identification of any specimens used in such assessment is critically important to reach correct conclusions. The authors should themselves be primarily responsible for the quality of the data they publish, and as we argue in the section ‘Need for Taxonomic Validation of Museum Specimens’ taking identifications of specimens in natural history collections at face value is unwise. The statement by Meiklejohn et al. that they used “curated reference material” is misleading, although undoubtedly not intentionally so. A specimen being placed in the unit tray for a particular species is no guarantee that the specimen is reliably identified. If the authors of the original study lacked entomological expertise, we wonder why they did not collaborate with entomologists (such as curators at the USNM). A closer collaboration with entomological experts would certainly have revealed the major flaw in the study design of Meiklejohn et al. – analyzing single specimens per species, many of them from known species complexes, without expert confirmation of the identifications – during the data acquisition and analysis phase instead of post publication. 

10) Line 128-149: This reviewer agrees that identification of biological materials for forensics should be tackled interdisciplinary and with common sense, by looking at geographic information, etc. However, it is incorrect to suggest that Meiklejohn et al. scope was discussing forensics and DNA barcodes as their paper was solely on the utility of the DNA barcoding databases using a subset of curated museum species. It is unjust to assume the co-authors of Meiklejohn et al wrote in terms of forensics because of their affiliation. Presumably, Meiklejohn et al. would have been more specific regarding cautions if their objective was for forensic application. However, this was not the case. The authors need to modify the text to reflect the inaccuracy. 

Pentinsaari et al: Considering the affiliation of the authors of the original study, it is certain the Meiklejohn et al. publication was widely viewed by the forensic science community and provided it with an unfounded pessimistic picture of DNA barcoding. Our manuscript seeks, in part, to correct this through a follow-up study. The authors also state that they selected some of their target species based on forensic importance. However, the reviewer is correct in stating that the original study was not specifically addressing the application of DNA barcoding in forensics. We have modified the title of our manuscript and changed the wording referring to forensics in other parts of the text.

11) Line 160-166: Authors briefly summarized how it is possible to use GPS coordinate metadata included in BOLD to better source and identify samples. This is a reasonable suggestion for those who wish to more accurately identify a species for purely taxonomic reasons. However, doing so would require subject expertise in entomology or arthropod evolution to properly utilize this type of data (e.g., specimen images and GPS coordinates) which is not reasonable to expect for most users of BOLD whose background knowledge will vary wildly from one user to the next. Finally, this kind of analysis was outside the scope of work performed by Meiklejohn et al. and therefore an unreasonable criticism of their methodology. This section either needs to be removed, or placed within context of the goal/purpose of the Meiklejohn et al study. 

Pentinsaari et al: This section does not criticize the study by Meiklejohn et al., but instead discusses the connection between BOLD and GenBank and differences between them which might not be familiar to many readers. The set of species analyzed by Meiklejohn et al. is only used as a convenient example to present these differences. Reporting all relevant metadata for natural history specimens is good practice even if those data are not essential for the study as such metadata can improve the reproducibility/interpretability of data and can be highly useful in subsequent analyses or validation of the data. For example, in the context of this study, the geographic origin of the Gryllus specimen (not reported by Meiklejohn et al.) reveals that it is highly unlikely to represent G. assimilis (see our reply to points 6-7 above).

12) Line 168-171: This statement overstates that 11 species did not escape operational errors. The sequences for only 3 species were able to be fully resolved, but the sequences for the other 8 species were not. Authors failed to meet burden of proof as well as contradicted themselves throughout the paper writing that these species were “not as straightforward” and should be viewed as “unconfirmed”.

Pentinsaari et al: Based on our reanalysis of the sequence data and subsequent examination of those specimens at USNM which could be traced and examined, we see no reason to change our conclusions regarding operational errors in the original study by Meiklejohn et al.

Minor Technical Comments:

1) Line 55-59: The impression the authors made here is that Meiklejohn et al used old specimens and thus this contributed to their misidentification results. Meiklejohn et al. included the age of the specimens as metadata and highlighted that amplicon/sequence length recovery was not correlated with specimen age. Thus, the authors need to modify the text to include this fact.

2) Line 80: Awkward wording of sentence. “Both BOLD and GenBank delivered generic identifications deemed incorrect for four specimens.” Not sure what this sentence means.

3) Line 155-158: Authors explain how BOLD and GenBank are similar. This is also mentioned in Meiklejohn et al and could have been acknowledged.

4) Line 175-178: Authors suggest that Meiklejohn et al. did not use proper controls and quality checks; however, they did this according to their protocols that was cited in Methods. Meiklejohn et al out-referenced these steps from their earlier paper (Meiklejohn et al 2018 International Journal of Legal Medicine). The authors need to update the text to reflect this. 

Comments on Tables and Figures:

Table 1: When checking the matches returned from the sequences listed in Table 1, it is apparent the authors mislabeled the query sequences. For example, upon downloading MK905402 from GenBank and searching against BOLD one returns a match for Forifucla auricularia. Thus, the authors need to place the red results from Meiklejohn et al under the MK905402/RC row for BOLD_all, not the MK905402 row alone. 

 Lastly, this table shows that 7 of 11 needed to be RC. However, only 3 species were able to be resolved. All 7 sequences that were not RC (as Meiklejohn et al. stated in their Discussion) were able to be confidently known to be incorrect due to their similarity statistic (<60%). Furthermore, the sequence orientation for BOLD was not explicated stated at the time Meiklejohn et al. queried the sequences and seems to only be affecting insect sequences (the BOLD fungal and plant algorithm was not sensitive to sequence orientation). 

Pentinsaari et al: Table 1 provides a direct comparison between the results reported by Meiklejohn et al. and our attempt to replicate these results. As a result, we include the incorrect matches obtained by Meiklejohn et al. on the BOLD_all row as originally reported instead of the reverse complement row. Any sequences uploaded to GenBank by Meiklejohn et al. as reverse complements have been converted by GenBank into the proper orientation, and therefore they now produce different query results than those reported in the original study. Our initial inability to reproduce the original results and subsequent discovery of what caused the very incorrect identifications for some insect specimens was a major motivator for compiling this manuscript. The lack of warning on BOLD about sequence orientation is addressed above in our response to comment number 3.

Table 2: What is N.D? the table description states “three categories of operational errors which compromised efforts by Meiklejohn et al…” Given this reviewer has concerns with the claims (and contradictions) the authors made about contamination, the ‘contamination’ column in this table needs to be removed. For instance, in line 85-87, the authors admitted that “the other three mismatches.. along the 17 tested species ruling out internal contamination.” Therefore, these species should not be included in the list as “contaminated” in the table. 

 Moreover, it is unclear what comprises the third operational error. Is it incorrect identification? Incorrect identification is not due to Meiklejohn et al. as these authors also had the same results. Neither Meiklejohn et al nor the authors performed taxonomic identification of the specimens. As presented, it appears to this reviewer that the authors wanted to find any reason for misidentification, aside from possible issues with BOLD. The column title needs to be modified accordingly.

Pentinsaari et al: Internal cross-contamination (i.e. between specimens included in the analyses of Meiklejohn et al.) could be ruled out for those three specimens, but as detailed above (reply to point 4) and in the manuscript, contamination with non-target DNA still explains the distant matches obtained beyond any reasonable doubt. While preparing the first draft of our manuscript, we did not examine the USNM loan records related to the Meiklejohn et al. material, the analyzed specimens, or their label data, but we did so during the review process. These studies have confirmed our original conclusions regarding the ambiguity in the identification of some of the analyzed specimens.

Figure 2: This is a great figure to show proper workflow. Meiklejohn et al. followed many of these features (e.g., negative controls, quality assessment, etc.). Also given their sequences were submitted to GenBank and are from a coding region, the sequences would also have been checked for premature stop codons, indels, etc (this is part of GenBank’s QC process before acceptance).

Additional Comments:

Appropriateness of the Methods

 There has been a shift in the guidelines for using BOLD for COI identifications – the interface now states that sequences must be submitted in forward orientation. The authors need to highlight that they searched the BOLD when this requirement had been articulated. 

 To support the claims made in this paper specifically on misidentification and cross contamination, the authors should have gone back to the pinned vouchered material at USNM to a) taxonomically confirm the species (addressing misidentification errors), and b) independently resequencing the questionable specimens (addressing cross-contamination issues). The addition of these steps would provide the authors evidence to classify the errors reported by Meiklejohn et al. A description of this work was not given.

Support of the Conclusions by the Results

 The authors identified three reasons for the misidentification reported in Meiklejohn et al: 1) reverse complement sequences were searched against BOLD, 2) sequences were cross-contaminated, and 3) taxa included were either misidentified or part of a species complex. The authors results supported conclusion 1), however only circumstantial evidence/results wre provided to support conclusions 2 and 3. 

Over reaching Conclusions

Several conclusions appear overreaching, especially those on misidentifications and cross-contamination. To validate these conclusions as previously stated, the authors should have gone back to the pinned vouchered material at USNM repository to a) taxonomically confirm the species (validating whether the specimens initially tested were indeed misidentified), and b) independently resequencing from questionable specimens (addressing cross-contamination issues).

Pentinsaari et al: Since all the cases we have listed as cross-contaminations involve pairs of very distantly related taxa, they are easily identified as cross-contaminations through comparison of the Meiklejohn et al. data to other independently generated sequences of the target and non-target taxa. Resequencing the original specimens would not add any value to our study. The comments related to controversial identifications of some of the analyzed specimens and changes in the BOLD interface have been addressed earlier in this response.

---

## [Editor Report · Decision Letter 2]

2 Apr 2020

BOLD and GenBank revisited – do identification errors arise in the lab or in the sequence libraries?

PONE-D-19-20600R2

Dear Dr. Hebert,

We are pleased to inform you that your manuscript has been judged scientifically suitable for publication and will be formally accepted for publication once it complies with all outstanding technical requirements.

With kind regards,

Matjaž Kuntner

Academic Editor

PLOS ONE

Additional Editor Comments (optional):

Thank you for your thorough rebuttal of the recent critique. Your paper's acceptance is long overdue.
---

## [Editor Report · Acceptance letter]

6 Apr 2020

PONE-D-19-20600R2 

BOLD and GenBank revisited – do identification errors arise in the lab or in the sequence libraries? 

Dear Dr. Hebert:

I am pleased to inform you that your manuscript has been deemed suitable for publication in PLOS ONE. Congratulations! Your manuscript is now with our production department. 

With kind regards,

on behalf of

Dr. Matjaž Kuntner 

Academic Editor

PLOS ONE